# R-loops and regulatory changes in chronologically ageing fission yeast cells drive non-random patterns of genome rearrangements

**David A. Ellis**[1¤], **Félix Reyes-Martín**[2], **María Rodríguez-López**[1], **Cristina Cotobal**[1], **Xi-Ming Sun**[3,4], **Quentin Saintain**[1], **Daniel C. Jeffares**[1¤], **Samuel Marguerat**[3,4], **Víctor A. Tallada**[2], **Jürg Bähler**[1]*

**1** Institute of Healthy Ageing, Department of Genetics, Evolution & Environment, University College London, London, United Kingdom, **2** Centro Andaluz de Biología del Desarrollo, Universidad Pablo de Olavide/Consejo Superior de Investigaciones Científicas, Seville, Spain, **3** MRC London Institute of Medical Sciences, London, United Kingdom, **4** Institute of Clinical Sciences, Imperial College London, London, United Kingdom

¤ Current address: Ear Institute, University College London, London, United Kingdom (D.A.E.); Department of Biology, University of York, York, United Kingdom (D.C.J)

\* j.bahler@ucl.ac.uk

**Data Availability Statement:** The genome-sequence, ChIP-seq and DRIP-seq datasets generated and analysed during the current study

## Abstract

Aberrant repair of DNA double-strand breaks can recombine distant chromosomal breakpoints. Chromosomal rearrangements compromise genome function and are a hallmark of ageing. Rearrangements are challenging to detect in non-dividing cell populations, because they reflect individually rare, heterogeneous events. The genomic distribution of *de novo* rearrangements in non-dividing cells, and their dynamics during ageing, remain therefore poorly characterized. Studies of genomic instability during ageing have focussed on mitochondrial DNA, small genetic variants, or proliferating cells. To characterize genome rearrangements during cellular ageing in non-dividing cells, we interrogated a single diagnostic measure, DNA breakpoint junctions, using *Schizosaccharomyces pombe* as a model system. Aberrant DNA junctions that accumulated with age were associated with microhomology sequences and R-loops. Global hotspots for age-associated breakpoint formation were evident near telomeric genes and linked to remote breakpoints elsewhere in the genome, including the mitochondrial chromosome. Formation of breakpoint junctions at global hotspots was inhibited by the Sir2 histone deacetylase and might be triggered by an age-dependent de-repression of chromatin silencing. An unexpected mechanism of genomic instability may cause more local hotspots: age-associated reduction in an RNA-binding protein triggering R-loops at target loci. This result suggests that biological processes other than transcription or replication can drive genome rearrangements. Notably, we detected similar signatures of genome rearrangements that accumulated in old brain cells of humans. These findings provide insights into the unique patterns and possible mechanisms of genome rearrangements in non-dividing cells, which can be promoted by ageing-related changes in gene-regulatory proteins.

are available in the European Nucleotide Archive repository under the study accession PRJEB30570. The RIP-chip data are available in ArrayExpress under accession number E-MTAB-7618.

**Funding:** This research was funded by a BBSRC-DTP studentship to DAE (London Interdisciplinary Doctoral Programme) [grant number BB/M009513/1] funded by the Biotechnology and Biological Sciences Research Council (https://bbsrc.ukri.org/); a Wellcome Senior Investigator Award to JB [grant number 095598/Z/11/Z] funded by the Wellcome Trust (https://wellcome.org/); and Medical Research Council funding to SM (https://mrc.ukri.org/). The funders had no role in study design, data collection and analysis, decision to publish, or preparation of the manuscript.

**Competing interests:** The authors have declared that no competing interests exist.

## Author summary

DNA breaks followed by chromosomal rearrangements that join non-neighboring DNA sequences may critically affect gene function, evolution, and ageing. Such chromosomal rearrangements are difficult to spot in sequence data even if they are widespread, because they are individually rare and reflect diverse events. Here we establish sensitive analyses of DNA sequences and identify prevalent rearrangements that specifically accumulate during ageing in yeast cells. These rearrangements feature short repeated DNA sequences near the breaks, preferentially occur in certain locations of the chromosomes (e.g., near their ends), and can link sequences originating from different chromosomes. We show results indicating that DNA-RNA interactions, triggered by the ageing-associated suppression of an RNA-binding protein, can cause the non-random patterns of some chromosomal rearrangements. Our analyses suggest that similar patterns of chromosomal rearrangements accumulate in brain cells in older humans, raising the possibility that such DNA changes occurring in ageing cells are conserved from yeast to human.

## Introduction

Cellular processes like transcription and replication can trigger DNA lesions such as double-strand breaks (DSBs) [1–4]. A sensitive sequencing approach has revealed DSBs at hotspots in mouse brain cells, linked to transcribed genes with neuronal functions [5], suggesting that the physiological context can affect the landscape of DSBs. DSBs are normally repaired by homologous recombination or by non-homologous end-joining (NHEJ), two pathways which protect chromosomes from aberrant structural variations [6–8]. Under certain physiological conditions, e.g. when the regular DNA-repair pathways are compromised, alternate DNA end-joining processes take over, often involving short homologous sequences (microhomologies) that are typically unmasked through DNA-end resection from the DSBs [9–11]. Microhomology-mediated end-joining (MMEJ) can link chromosomal breakpoints that are normally far apart or even on different chromosomes [12,13]. Such events lead to genome rearrangements such as inversions, duplications, translocations or deletions, which can considerably affect the function of genomes. The patterns of genome rearrangements are shaped by the particular mechanisms of their formation and by the fitness effects they exert on the cell.

Ageing has been associated with both an increase in DSBs [14,15] and a decline in the efficiency and accuracy of DNA repair [15,16]. Accordingly, increased genomic instability and chromosomal rearrangements are well-known hallmarks of ageing [17–22]. Impaired NHEJ leads to accelerated ageing in human patients and mouse models, and MMEJ increases with age [23]. Genome re-sequencing studies during ageing have been limited to mitochondrial DNA [24,25], small genetic variants [26,27] and duplications [28], or proliferating cells [29]. No systematic approaches have been applied to identify heterogeneous, rare chromosomal rearrangements in non-dividing, somatic cells [15,30–32].

Processes affecting ageing are remarkably conserved from yeast to human, including both genetic and environmental factors [21,33]. The fission yeast, *Schizosaccharomyces pombe*, is a potent model for cellular ageing; we and others have explored effects of nutrient limitation, signalling pathways and genetic variations on chronological lifespan in *S. pombe* [34–37]. Chronological lifespan is defined as the time a cell survives in a quiescent, non-dividing state, which models post-mitotic ageing of somatic metazoan cells [21,33]. Quiescent *S. pombe* cells feature distinct DNA-damage responses [28,38,39] and distinct mutational forces that can

promote genetic diversity [26,27]. Here we interrogate aberrant genomic DNA-junction sequences in non-dividing *S. pombe* cells, revealing unique signatures of ageing-associated chromosomal rearrangements and suggesting their mechanistic underpinning. Similar patterns of rearrangements are also evident in ageing human brain cells.

## Results and discussion

### Microhomology-associated genome rearrangements specifically increase in ageing yeast cells

We previously sequenced eight chronologically ageing pools of *S. pombe* cells, generated from advanced intercross lines of strains Y0036 [40] and DY8531 [41], and analysed changes in the standing genetic variation as a function of age to identify longevity-associated quantitative trait loci [34] (Materials and Methods). Here we report the striking new structural variation that arose in these cellular pools during ageing. Structural variant calling software usually requires support from multiple sequence reads [42–51]. Whilst this is useful for reducing false positives, these algorithms will only identify variations present in multiple cells in a population. To identify the rare, heterogeneous variations arising spontaneously in different non-dividing cells, we stringently filtered split reads that joined sequences from two distant genomic sites (S2 Fig) [34]. Such split reads represent potential breakpoint junctions of genome rearrangements that lead to new sequence combinations (Fig 1A). Examples of split reads are provided in S1 Fig. We identified 776,174 such junctions from 225,554,047 total reads across eight replicate pools sampled at six time points.

Several lines of evidence indicate that these breakpoint junctions are not artefacts of sequence library preparation but represent *in vivo* genome rearrangements. First, fewer junctions were present within coding regions than would be expected by chance (Fig 1B). This bias may reflect selection against intra-genic rearrangements that disrupt gene function. Second, modelling showed that the free DNA ends available for junction formation were not proportionally represented in the observed juxtapositions, i.e. sequences represented by higher read depth were not more likely to feature in breakpoint junctions (S3 Fig). Third, a larger age-associated increase was evident in intra-chromosomal junctions than in inter-chromosomal junctions, and among intra-chromosomal events, junctions joining neighbouring regions were preferred over those joining more distal regions (S4 Fig; S1 Text). Such bias is not expected from a bioinformatics artefact. Fourth, age-associated junctions were characterised by separate repair signatures at breakpoints compared to signatures suggestive of false positives, of which there were far fewer (see below and S1 Text). A drawback of this approach is that the juxtaposed regions forming a junction are analysed without supporting information from other reads or breakpoints, and the exact nature and extent of the structural variations thus remains unknown.

During chronological ageing, breakpoint junctions strongly increased relative to the total number of mapped reads in each sample, particularly from Day 2 onwards (Fig 1C). This increase was most pronounced for junctions involving nuclear DNA only, but was also evident within mitochondrial DNA and between nuclear and mitochondrial DNA (Fig 1C). Other work supports the notion that DSBs in nuclear DNA can be repaired with mitochondrial DNA [29,52,53]. The breakpoint junctions featured different types of sequence rearrangements: single-base insertions not present at either joined region, blunt junctions directly linking two regions, or microhomologies of up to 20 bases shared between both joined regions (Fig 1D). The blunt junctions and the junctions with single-base insertions or single-base microhomologies did not increase with age. These junctions might have been formed by a distinct mechanism before ageing and/or they could reflect artefacts (S1 Text). In stark contrast, junctions

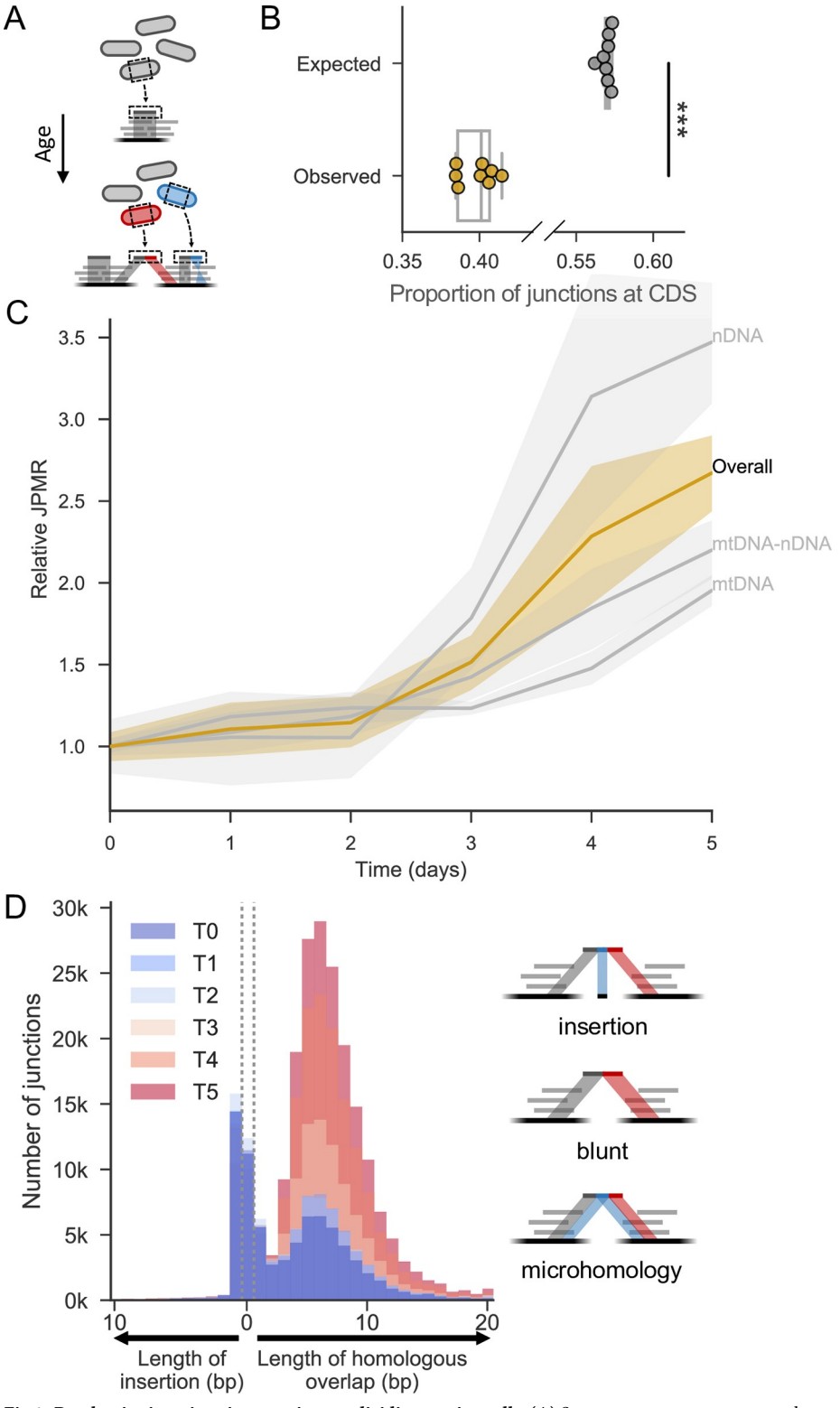

**Fig 1. Breakpoint junctions increase in non-dividing, ageing cells. (A)** Spontaneous rearrangements that occur in non-dividing cells should be heterogeneously distributed throughout the population and identifiable in DNA sequencing data. **(B)** Proportion of junctions expected to fall in coding regions *vs* proportion that do (based on simulated data; Materials and Methods). Points represent biological (yellow) or *in-silico* (grey) repeats. One-sided two sample Mann-Whitney U to test whether observed is less than expected (U = 0, p <0.001, N = 8). Note that proportion

of simulated junctions in coding regions is similar to proportion of genome reported to be coding (excluding introns [59]). **(C)** Number of breakpoint Junctions Per Mapped Read (JPMR) passing filter (Materials and Methods) as a function of cellular age. Cells were cultured in rich medium until the optical density no longer increased, indicating that cells were no longer dividing, and used as Day zero for the chronological ageing timecourse. Samples were taken from the cultures at Days 0, 1, 2, 3, 4 and 5. JPMR are shown relative to Day zero (N = 8, confidence interval = 68%). Ochre line: total junctions, genome wide. Grey lines from top to bottom: subset of junctions formed among fragments of nuclear DNA (nDNA); between fragments of mitochondrial (mtDNA) and nDNA; among fragments of mtDNA. **(D)** Histograms showing number of junctions with various lengths of microhomology (right of dotted lines), blunt joints (between dotted lines), or non-homologous insertion (left of dotted lines) at the breakpoint and during cellular ageing. Histograms at different days are overlaid on top of each other, with early timepoints in blue through to late timepoints in red (see colour legend). Right: schemes for reads with each type of junction.

featuring 2–20 bases of microhomology did markedly accumulate with age (Fig 1D). This signature indicates that these ageing-associated rearrangements occur by MMEJ. The observed size distribution, with a peak at 5–6 bases, might reflect a trade-off between the length of microhomology available near DSBs and the benefit of longer homology for end-joining repair. Interestingly, rearrangements with similar patterns of microhomology seem also to be enriched in cancer cells [54]. We conclude that genome-wide rearrangements, represented by breakpoint junctions featuring microhomologies, accumulate as a function of the age of non-dividing cells.

Using motif discovery, we found seven long sequence motifs enriched at microhomology-mediated junctions (S5A Fig). Motifs of known transcription factors from fission and budding yeast, which are shorter than the longer sequence motifs discovered, showed significant homology within these longer motifs (S5B Fig). These transcription factors are involved in nutrient starvation and other stress responses or in cell-cycle control. This result suggests that specific transcription factors are associated with regions near chromosomal junctions, raising the possibility that they are involved in triggering DSBs and microhomology-mediated rearrangements.

## Similar patterns of genome rearrangements accumulate in old human brain cells

A recent study, looking at single-nucleotide polymorphisms in single cells, reports that somatic mutations accumulate with age in humans [55]. Non-dividing yeast cells are a model for the post-mitotic ageing of somatic human cells such as the long-lived cells of the brain [33,56]. To check whether similar rearrangements also occur in human cells, and to validate our method in an independent system, we applied our junction calling pipeline to published sequencing data of young and old adult brain tissue [24]. We found a subtle increase in junctions in older brain cells (Fig 2A), although differences were marginally significant at best, reflecting that the coverage and sample number in this data set were low (Materials and Methods). Note that the younger brains in this analysis will have already experienced some level of age-associated physiological decline. The most striking result from this analysis was the similar pattern of rearrangements: as in fission yeast (Fig 1D), the junctions were associated with microhomology in somatic human cells (Fig 2A). The microhomology-associated junctions in human brain cells showed a bimodal distribution: a large population featuring similar microhomology lengths to fission yeast (peaking around 4–6 bases), and a less abundant population featuring longer microhomology (median ~16 bases). Interestingly, in cancer genome sequences a transition in the probability of junction formation occurs at around 11 bases of microhomology [54]; the authors suggest that this transition reflects a shift in repair mechanisms from MMEJ to single-strand annealing. Notably, simulated data showed that there were fewer junctions in coding regions of human brain cells than would be expected by chance (Fig 2B). As for fission yeast

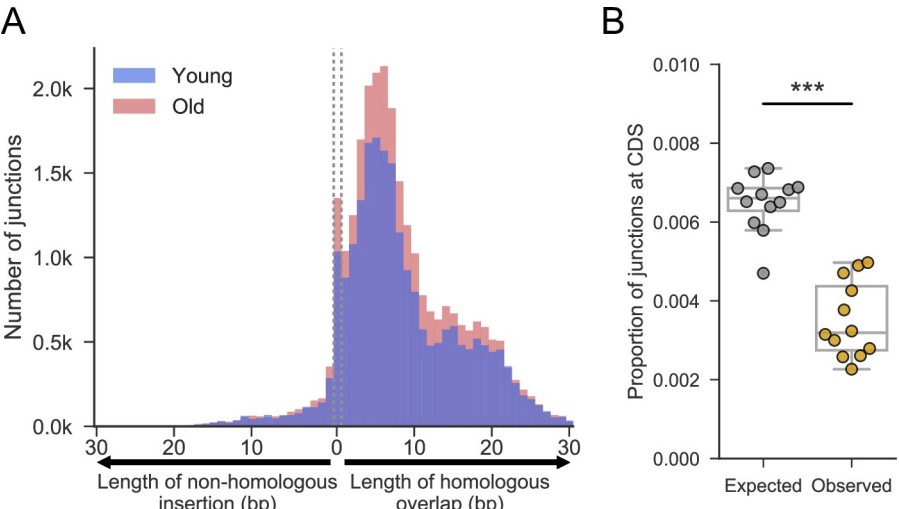

**Fig 2. Similar patterns of genome rearrangement in ageing human brain cells as in ageing yeast cells.** Analysis was performed on sequence data from putamen samples of neurologically normal Caucasian males [24], six of whom were young (blue; 19, 24, 28, 28, 29 and 30 year old donors) and six of whom were old (red; 67, 71, 78, 83, 85 and 89 year old donors). The 100bp reads from these samples, which were crudely enriched for mtDNA, were re-mapped to the whole human genome and filtered (Materials and Methods). **(A)** Histograms showing number of junctions with various lengths of microhomology (right of dotted lines), blunt joints (between dotted lines), or non-homologous insertions (left of dotted lines) at breakpoint. Histograms at different ages are overlaid on top of each other, with samples from young donors in blue and older donors in red (as in A). In these sparse data, the difference between young and old brain tissue was marginally significant at best (comparison of number of junctions per mapped read: $p_{Mann-Whitney}$ = 0.23; correlation between sample age and JPMR: $p_{Pearson}$ = 0.17; comparison of average microhomology length: $p_{T-test}$ = 0.05, suggesting slightly shorter mean microhomologies in old tissues, 8.15bp *vs* 8.28bp). **(B)** Proportion of junctions expected to fall in coding regions *vs* proportion that do (based on simulated data; Materials and Methods). One-sided two sample Mann-Whitney U to test whether observed is less than expected (U = 3, p <0.001). Simulated proportions approached the 1.5% proportion of the genome reported as coding [127] when more junctions were simulated.

(Fig 1B), this finding likely reflects selection against rearrangements that interfere with gene function, either through cell death or active culling of unfit cells [57]. These results raise the possibility that similar patterns of ageing-related DNA rearrangements occur in both yeast and human brain cells.

## Local and global hotspots for genome rearrangements in ageing yeast cells

Junction formation over time represents a complex, multi-dimensional process: each junction is comprised of two juxtaposed sequences from any two genomic regions; independent junctions can recurrently form between the same two regions (Fig 3A, green), or between one 'hotspot' region and different other regions (Fig 3A, blue & red); and junction-formation can be either age-dependent (Fig 3A, red) or not (Fig 3A, blue). To visualize the yeast junctions and identify age-associated patterns in any rearrangements, we determined ratios of the number of junctions in the oldest cells to the corresponding number in the youngest cells. Ageing-associated junctions formed preferentially at two distinct types of hotspot: 1) those enriched for local, intra-chromosomal junctions (mostly within 20kb), and 2) those enriched for more global, inter-chromosomal junctions (Fig 3B). The local hotspots were more abundant but less pronounced than the global hotspots (Fig 3B). These local hotspots likely reflect spatial constraints on junction formation, with neighboring DNA being a more likely repair substrate than distal DNA (S4 Fig). At global hotspots, on the other hand, junctions between nearby regions of DNA were under-represented, contrary to expectation and the situation at local hotspots (S4 Fig). These findings suggest that distinct mechanisms operate at global and local hotspots.

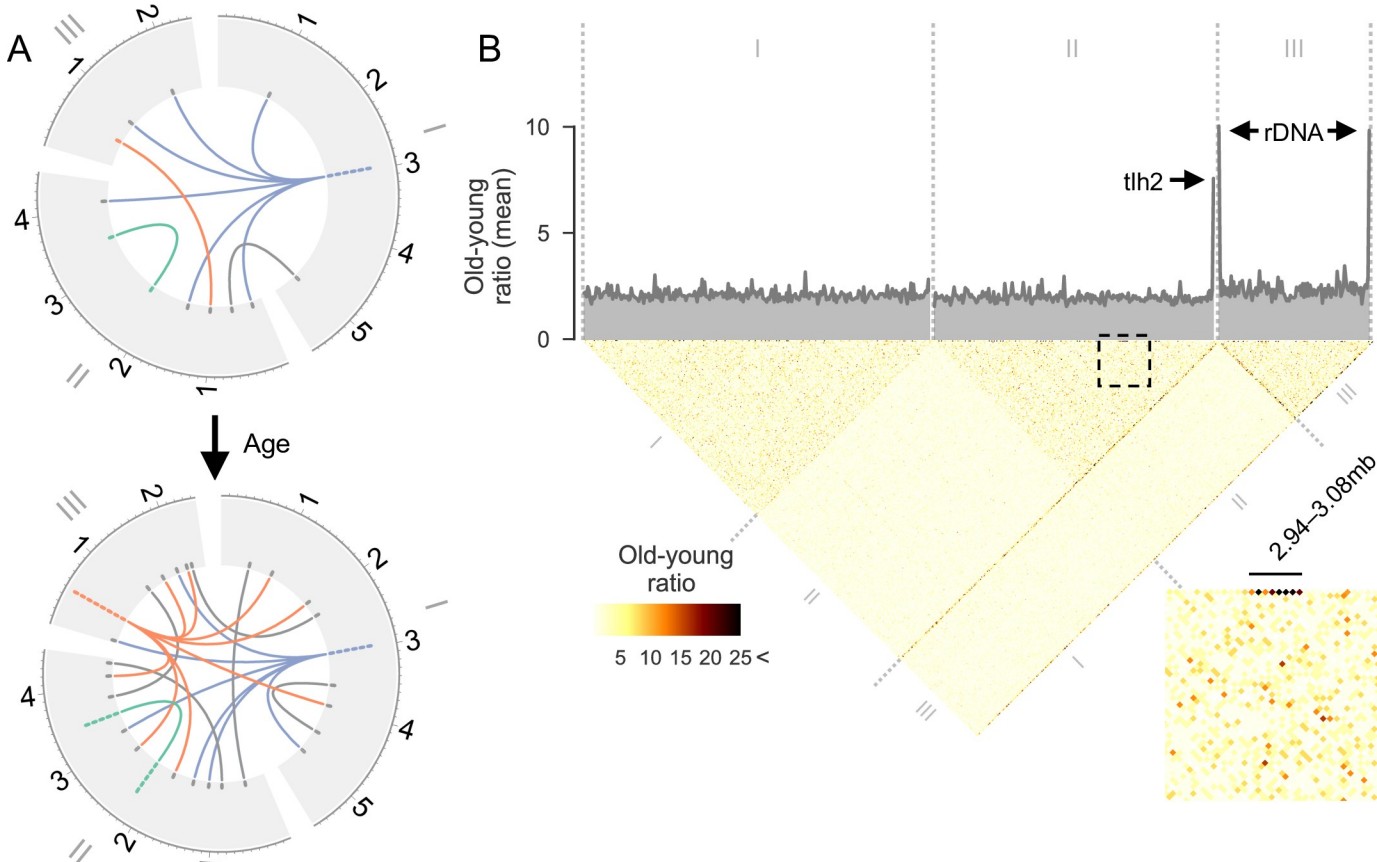

**Fig 3. Local and global hotspots for age-associated junction formation.** (A) Illustration of complexity of junction formation over time. In this example, junctions are represented as a link between two points in any of the three *S. pombe* chromosomes (depicted around the edge in Mbp). The number of junctions in a region are stacked up in the outer ring. Blue: although a region may have many junctions at a late time point (bottom), these junctions may not be the result of ageing as they could have been present earlier (top). Red: true age-associated hotspots will show an age-associated increase. Some rearrangements may occur recurrently between the same two regions (green) or one region with many others (red). **(B)** Heatmap showing the ratio of junction counts between Days 5 and 0 (T5/T0; 'old-young ratio'), in 20kb windows across the genome. An element in the heatmap depicting two regions with high levels of age-associated breakpoint formation will have a higher old-young ratio and appear darker. The average number of breakpoints at each window, genome-wide, is plotted in grey at the top. An intra-chromosomal region of Chromosome II is blown up at bottom right; local hotspots are exemplified by Region II:2940000–3080000, where the old-young ratio exceeds 25 in many bins along the top edge of the heatmap. Global hotspots are reflected by the dark diagonal emanating from the right end of Chromosome II and both ends of Chromosome III.

We identified three strong global hotspots featuring numerous connections with other, typically remote sequences throughout the genome. These global hotspots were located near the right end of Chromosome II and near both ends of Chromosome III, the latter being the sites of ribosomal DNA (rDNA) (Fig 3B). Given that these global hotspots occur in repetitive regions near chromosome ends [58–60], they might simply reflect the large number of repeated sequences. However, if the copy number of these repeated sequences remained constant during ageing, the ratio of junctions between young and old cells should still reflect ageing-associated changes (S6A Fig). We therefore checked for changes in the ratio of repeat copy numbers between young and old cells at global hotspots. This analysis showed that repeat sequences at hotspots did not increase, but actually decreased with age (S6B Fig). Thus, if anything, we under-estimated the prevalence of the global hotspots as sites for age-associated junction formation. Work in other systems has shown that copy numbers of rDNA repeats decrease with age, and instability in this region is linked with ageing and longevity [61–63]. The decrease of rDNA copy numbers with age could be linked to junction formation in these

regions: work in primate kidney cells has demonstrated differential repair of cellular DNA sequences [64], reminiscent of the heterogeneity we observe for age-associated rearrangements. We conclude that ageing-associated rearrangements occur preferentially at either local or global hotspots.

## Possible causes for global hotspots downstream of *tlh2*

To better understand the global hotspots, we analysed the positions of junctions relative to genome annotations. The first hotspot was downstream of a *tlh2* gene copy (Fig 4A; left), encoding a RecQ family DNA helicase. Copies of *tlh2* reside on all four sub-telomeres of Chromosomes I and II [60]. The *S. pombe* reference genome assembly only includes one *tlh2* gene copy, and the other three copies may also feature hotspots. Initially discovered in bacteria [65], RecQ helicases are highly conserved. Notably, mutations in two human RecQ helicases, BLM and WRN, lead to premature ageing through Bloom or Werner syndromes [66] and another, RECQL5, alleviates transcriptional stress [67]. The other two hotspots were near the 5.8S, 18S and 28S ribosomal RNA genes (Fig 4A; middle and right). At all hotspots, junctions were enriched at the 3'-ends of these genes (Fig 4A). Ribosomal RNA genes are highly transcribed and common sites of transcription and replication stress [1,3]. Moreover, rDNA repeats can become unstable during ageing and cause cell death [61–63].

A re-analysis of RNA-seq data from non-dividing cells [68] revealed a subtle increase in *tlh2* expression during chronological ageing (S7A Fig). To test whether increased transcription at *tlh2* leads to increased junction formation at this hotspot, we generated a strain overexpressing *tlh2*. We then sequenced the genome of this strain, along with a wild-type control, in early stationary phase when *tlh2* expression is normally low. The proportion of junctions downstream of *tlh2* was higher in the *tlh2* overexpression strain compared to wild-type (S7B Fig). Thus, overexpression of *tlh2* is sufficient to trigger increased breakpoint junctions, possibly reflecting rearrangements owing to transcriptional stress. Moreover, the *tlh2* overexpression strain was substantially shorter-lived than wild-type cells (S7C Fig). These results raise the possibility that increased transcription of *tlh2* and/or increased levels of the Tlh2 protein cause rearrangements that affect cell survival and longevity.

Given the subtle and variable increase in *tlh2* expression based on bulk RNA-seq data (S7A Fig), we wondered about other causes of this hotspot. Initially, we hypothesised that *tlh2* might be transcribed heterogeneously during ageing (variegation), with rearrangements being limited to a few cells featuring high *tlh2* expression. Single-molecule RNA fluorescence *in situ* hybridization (smFISH) experiments showed heterogeneous expression in cells overexpressing *tlh2*, but no *tlh2* expression was evident in wild-type cells during ageing (Fig 4B). Therefore, we inspected the available RNA-seq data [68] over a larger genomic region. This analysis revealed a pronounced RNA peak in the hotspot region downstream of *tlh2*, dwarfing the expression of *tlh2* itself, and this peak greatly increased in older cells (Fig 4C). Thus, the whole region shows age-dependent increases in transcription, with particularly pronounced changes downstream of *tlh2*. How might these transcriptional changes occur and trigger the hotspot? Below, we present results suggesting the involvement of chromatin alterations and R-loop formation.

Together with other chromatin remodelling factors, the *S. pombe* sirtuin Sir2 modulates *tlh2* expression by modifying chromatin at telomeres [69]. Accordingly, our smFISH experiment showed that *tlh2* was de-repressed in *sir2* deletion cells (Fig 4B). From work in budding yeast, Sir2 is implicated in Ku-dependent NHEJ of DSBs [70,71], and it prevents DNA damage in proliferating cells [72]. In chronologically ageing cells, however, Sir2 may actually promote DNA damage [73]. Intriguingly, the Ku-Sir2 complex normally binds to subtelomeres but

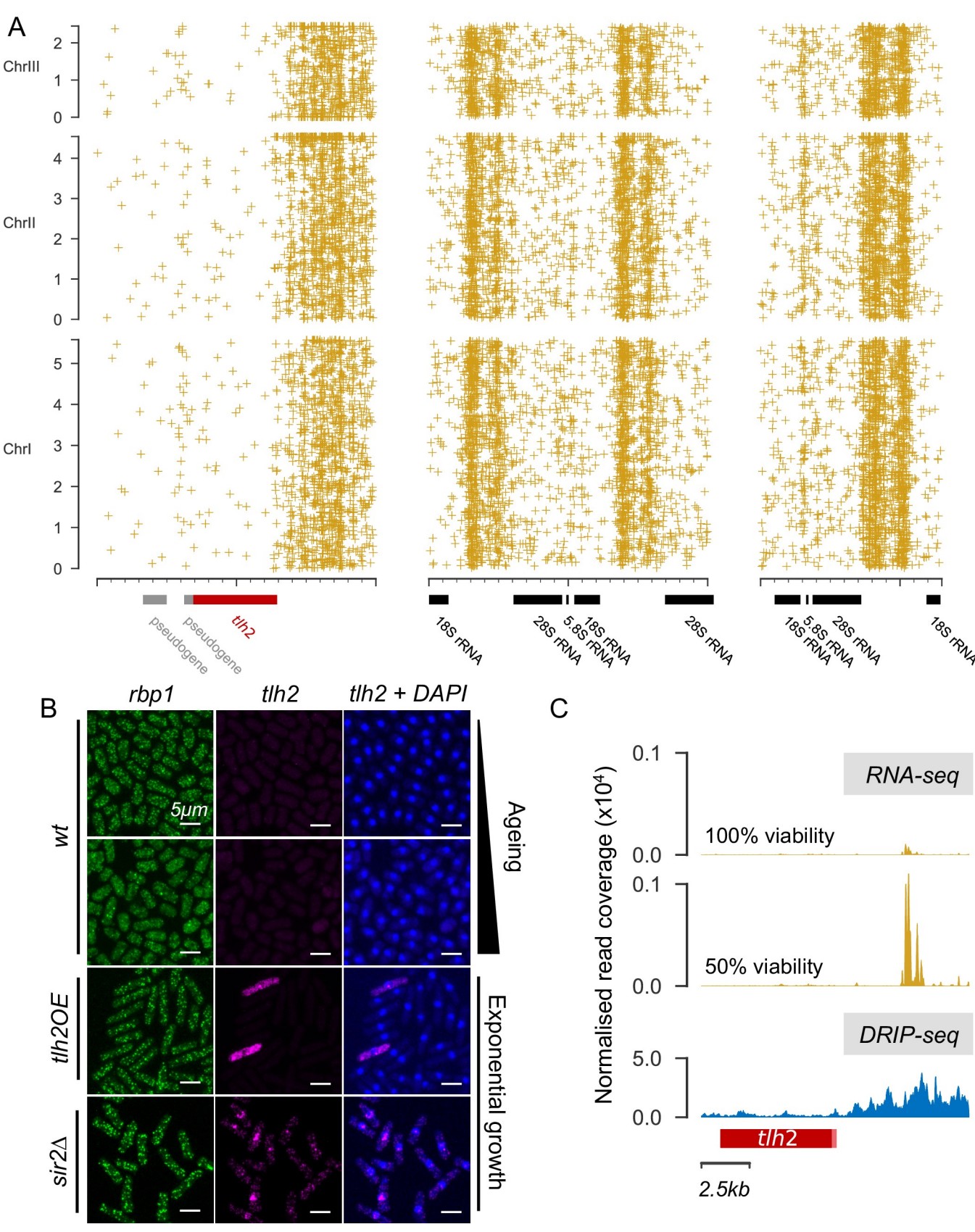

**Fig 4. Bodies of RNA associated with DNA at global hotspots. (A)** Scatter plot showing junctions at each of the global hotspots identified near telomeres (Fig 3). Coordinates of junctions in ~20kb region surrounding each hotspot are shown on x-axis, and coordinates of corresponding junctions in Chromosomes I-III are shown on y-axis. Annotations including pseudogenes (grey), *tlh2* (red) and rRNA genes (black) are shown beneath x-axis. **(B)** smFISH experiment using probes against the housekeeping control gene *rbp1* (green, left) and *tlh2* (purple, middle and right), with DAPI-stained nuclei (blue, right). Non-dividing wild-type (wt) cells do not appear to show any *tlh2* expression after 2 days (top row) or 4 days (second row) of chronological ageing, proliferating cells overexpressing *tlh2* (tlh2OE) show strong, but highly heterogeneous expression, while proliferating *sir2Δ* mutants show homogeneous high expression in all cells. **(C)** Top: RNA mapping to an unannotated region downstream of *tlh2* in young (100% cell viability) and old (50% cell viability) stationary-phase cells. Each sample's read coverage is normalised to the total number of mapped reads for that sample, showing the mean of two replicates (reanalysed from ref. 67). Bottom: DRIP-seq in proliferating cells using the αS9.6 antibody suggests that R-loops accumulate at the same region. The broad DRIP-seq signal points to the presence of multiple R-loops downstream of *tlh2*, reflected by the neighbouring peaks.

moves to DSBs as they occur elsewhere in the genome, which is associated with a loss of silencing at subtelomeres [74–76]. This mechanism suggests that with increasing age-associated genome damage, Sir2 and other NHEJ factors may move away from global hotspot regions, thus exposing them to rearrangement through MMEJ. To investigate this possibility, we analyzed non-dividing wild-type and *sir2Δ* cells of different age. Cells lacking Sir2 showed a subtle extension of chronological lifespan compared to wild-type, especially at later timepoints (Fig 5A). In budding yeast, the chronological lifespans of *sir2* deletion strains are variable and dependent on genetic background, leading to prolonged lifespan in some backgrounds and reduced lifespan in others [73]. Given the reported NHEJ-related function of Sir2 [70,71], we first assessed the age-dependent levels of insertions and deletions (indels), mutations which are associated with NHEJ. While wild-type cells showed a substantial increase of indels in aged cells, *sir2Δ* cells showed only a subtle increase (Fig 5B). This finding suggests that, as in budding yeast, Sir2 increases DNA damage in chronologically ageing cells. On the other hand, Sir2-dependent silencing might protect the genome, particularly at the *tlh2* hotspot region, from MMEJ-associated rearrangements in ageing cells. Indeed, while wild-type cells showed only a marginal accumulation of chromosomal junctions in aged cells, likely reflecting the slightly shorter time course compared to Fig 1C, *sir2Δ* cells showed an ~2-fold accumulation of junctions (Fig 5B). In particular, *sir2Δ* cells showed an age-dependent accumulation of junctions in the *tlh2* hotspot region (Fig 5C and 5D), including a strong accumulation over the region that showed an age-associated build-up of transcripts in Fig 4C. Taken together, these results support a model where an age-associated re-distribution of Sir2 de-represses chromatin at the global hotspot, thereby increasing transcription and genome rearrangements. Thus, although Sir2 may trigger age-dependent mutations associated with NHEJ, it may also offer protection from excessive MMEJ-associated rearrangements.

How could the transcriptional de-repression in the *tlh2* hotspot region trigger genome rearrangements in ageing cells? R-loops, RNA annealed to DNA, are common sites of genome instability: they can trigger DSBs through collisions with the transcription or replication machineries or through active processing by nucleotide excision-repair nucleases [4,77,78]; they can also directly interfere with DNA repair [79]. Notably, a screen in human cells for proteins binding to RNA/DNA hybrids has identified the LIG3 DNA ligase involved in MMEJ [80], raising the possibility of a mechanistic link between MMEJ and R-loops. Recent work in budding yeast has shown that sirtuins inhibit R-loop formation and associated genome instability, and cells lacking sirtuins show a genome-wide increase in nascent mRNAs [81]. Given the large accumulation of transcripts downstream of *tlh2* (Fig 4C), we wondered whether the global genome-rearrangement hotspot could be due to R-loops. To test whether the *tlh2* locus is prone to R-loop accumulation, we applied DNA-RNA immunoprecipitation followed by sequencing (DRIP-seq) for the profiling of R-loops [82]. Indeed, R-loops appeared to be strongly enriched downstream of *tlh2* in proliferating cells (Fig 4C). We also tried DRIP-seq in ageing cells, but these cells were too resistant to enzymatic or mechanical breaking. Despite

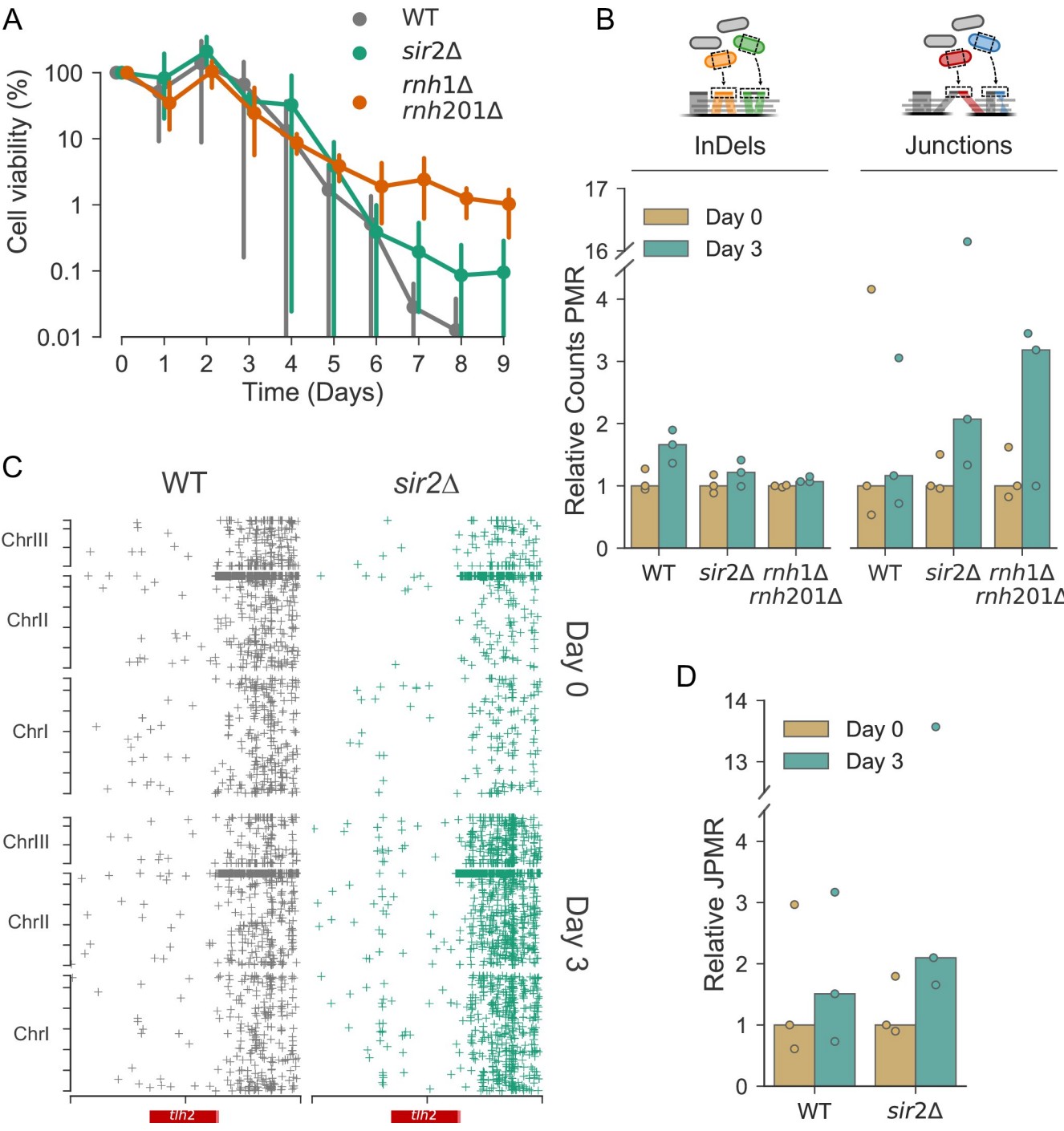

**Fig 5. Age-associated phenotypes in *sir2* and RNase H mutants.** (A) High-throughput lifespan assay of *rnh1Δ rnh201Δ* (orange), *sir2Δ* (green) and wild-type (grey) cells. Points show the mean of three independent repeats, and bars show the 95% confidence interval. Samples for sequencing were taken at Day 0 and Day 3, because too few live cells were present for sequencing analyses at later timepoints. (B) Counts of indels (left) and junctions (right) per mapped read (PMR), relative to Day 0, in wild-type, *sir2Δ* and *rnh1Δ rnh201Δ* cells. Dots show relative counts PMR for each replicate. (C) Scatter plot as in Fig 4A showing the coordinates of junctions at the hotspot surrounding *tlh2*. The x-axis shows the coordinates of junctions in 20kb region surrounding the *tlh2* hotspot (*tlh2* gene depicted in red); the y-axis shows the coordinates of the corresponding junctions across the genome. Junctions from three repeats were pooled, per sample, for wild-type samples (left, grey) and *sir2Δ* samples (right, green), both at Day 0 (top) and Day 3 (bottom). (D) Normalised data from (C). All junctions within the region were collected for each replicate and normalised to the number of mapped reads in that sample. Data is shown relative to the median at Day 0.

this caveat, these results raise the possibility that R-loops trigger the rearrangements at the global hotspot. These R-loops may be promoted by increased transcriptional activity downstream of *tlh2* (Fig 4C), triggered by ageing-dependent changes in chromatin-remodelling factors such as Sir2 (Fig 5B and 5C). This transcriptional activity could reflect small non-coding RNAs involved in chromatin regulation in this sub-telomeric region [83]. Overexpression of *tlh2* could exert its effects (S7 Fig) by affecting the remodelling of downstream chromatin or via an R-loop independent function of the Tlh2 protein.

## Ageing-related changes in RNA-binding proteins and R-loops may promote genome rearrangements at local hotspots

Given the bias towards the 3'-ends of genes at global hotspots, we analysed the position of all breakpoint junctions relative to coding regions (Materials and Methods). In agreement with Fig 1B, junctions were under-enriched in coding regions (Fig 6A). Moreover, significantly more junctions occurred in 3'-untranslated regions (UTRs) of genes than in 5'-UTRs, similar to the situation at the global hotspots. Our pipeline showed no bias towards AT- or GC-rich regions (Materials and Methods). This finding suggests that the ends of genes are particularly prone to rearrangements. Next, we identified genes whose 3'-UTRs contained more junctions than would be expected by chance, given their length (Fig 6B). Functional enrichment analysis using AnGeLi [84] showed that, of the 148 enriched genes, 17 produced transcripts that are targets of Scw1, representing a significant enrichment (FDR-corrected p <0.0001). Scw1 is an RNA-binding protein (RBP) [85–87] and an orthologue of the human polypyrimidine-tract-binding proteins RBPMS and RBPMS2.

Scw1 negatively regulates its target RNAs, possibly by binding to a motif in their 3'-UTRs [85]. To examine how an RBP might affect DNA rearrangements, we looked for any ageing-dependent changes in Scw1 substrate binding (Materials and Methods). These analyses suggested that Scw1 loses its affinity for some RNA targets with age (S8A Fig), but it does not switch binding substrate from RNA to DNA (S8B Fig). In fact, the protein levels of Scw1 markedly decreased in ageing cells (Fig 6C). The budding yeast orthologue of Scw1, Whi3, aggregates during replicative ageing [88], and Scw1 forms aggregates when overexpressed in fission yeast [89]. It is, therefore, possible that the decrease in Scw1 reflects protein aggregation. In any case, this result suggests that Scw1, rather than promoting rearrangements at its targets, is preventing them in young cells.

How might an age-associated functional loss of Scw1 trigger rearrangements? We hypothesized that R-loops might again be involved. The presence of RBPs on nascent transcripts can inhibit the formation of R-loops by preventing RNA from annealing to the single-stranded DNA template [90–93]. To test whether loss of Scw1 promotes R-loop formation, we quantified R-loops in *scw1* deletion (*scw1Δ*) and wild-type cells using R-loop immunostaining and fluorescence quantification. We validated the technique using RNase H mutants that are known to show increased R-loop formation as a positive control [94] and RNase H treatment as a negative control (S9 Fig). Moreover, almost all of the S9.6 signal was resistant to RNase III but sensitive to RNase H (S9C Fig), in contrast to results recently reported from whole cells [95], supporting the view that the antibody detected R-loops. For a physiologically relevant comparison, we used proliferating and early stationary phase cells, because older wild-type cells naturally lack Scw1 (Fig 6C). Indeed, the *scw1Δ* cells contained more R-loops than wild-type cells (Fig 6D). In a complementary, less artefact-prone approach, we applied DRIP-seq to directly test the effect of Scw1 on R-loop formation at the 227 published target genes of Scw1 [85]. This experiment showed that such R-loops slightly but significantly increased in the absence of Scw1 (Fig 6E). An increase remained even when the 17 Scw1 targets we identified

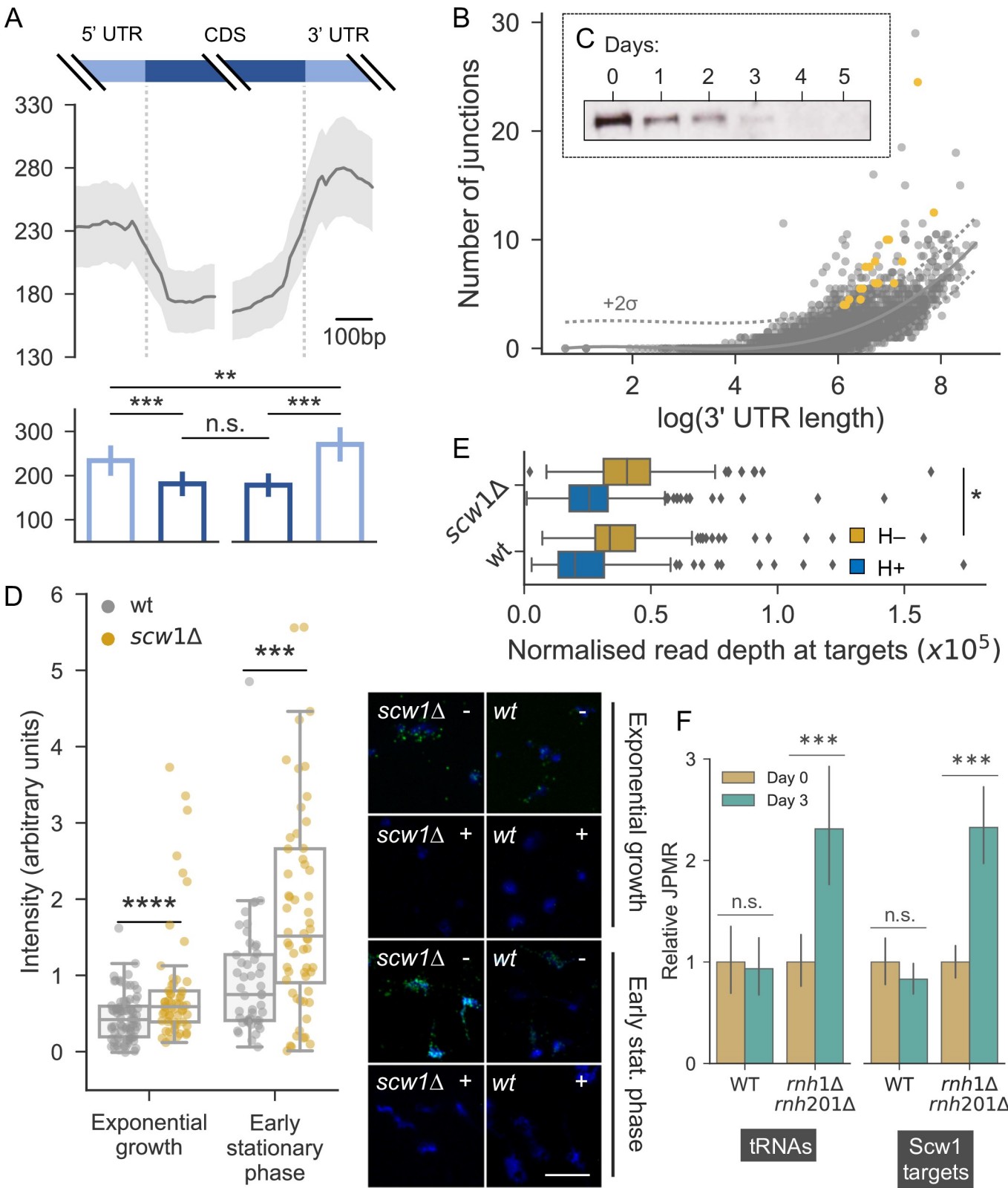

**Fig 6. R-loops may contribute to local hotspots in absence of Scw1. (A)** Top: sliding mean (10bp windows) junction counts around coding regions (CDS) across genome. Bottom: mean ± standard error of junction counts in 250bp bins on either side of coding regions across genome. Comparisons between bins were performed with paired t-tests (** p<0.001; *** p<0.0001; n.s., not significant). **(B)** Scatter plot of 3'-UTR length *vs* number of junctions in that UTR. Each

point represents a gene. Dotted lines show two standard deviations above population mean (solid line). All genes above dotted line were used for gene enrichment analysis. Scw1 targets >2x population standard deviation are in orange. (C) Western blot of Scw1-TAP at different days of cellular ageing. Coomasie blue was used as loading control. (D) Left: rapidly proliferating and early stationary phase cells (equivalent to Day 0 in C), which normally express Scw1, were analysed for R-loop formation in presence (wt) and absence (*scw1Δ*) of Scw1. Right: example images from chromosomal spreads; DNA stained with DAPI (blue) and R-loops by αS9.6 antibody (yellow), with (+) or without (-) RNase H treatment which removes R-loops. Note that almost all of the S9.6 signal co-localizes with DAPI, as seen with increased magnification and brightness to reveal faint DAPI signals. Scale bar: 10μm. (E) Normalised read coverage at Scw1 target genes from DRIP-seq of *scw1Δ* and wild-type (wt) cells using the αS9.6 antibody (ochre). The difference between strains is significant (one sample t-test: T = -2.52, p = 0.012). Samples after addition of RNase H are shown as control (blue). (F) Junctions per mapped read (JPMR), relative to Day 0, in wt and *rnh1Δ rnh201Δ* cells at tRNAs and core Scw1 target genes [85] as indicated. Bars show the mean (±95% confidence interval) JPMR at all tRNAs or Scw1 targets ±500bp. Days 0 and 3 were compared with paired t-tests for each genotype (tRNAs: wt T = 1.6 p = 0.11, rnh T = -4.8, p<0.0001; Scw1: wt T = 1.6, p = 0.12; rnh T = -6.4, p<0.0001). The wt likely showed no increase in junctions because chromosomal rearrangements normally only accumulate at later ageing timepoints (see Fig 1C).

(Fig 6B) were not included (one sample t-test: T = -2.46, p = 0.014). This finding suggests that the R-loops increased across all or most Scw1 targets. We conclude that the absence of Scw1 is sufficient for increased R-loop formation at its target genes.

To further test the role of R-loops in age-dependent genome rearrangements, we analysed the RNase H double mutant (*rnh1Δ rnh201Δ*), which is defective in R-loop processing [94]. The *rnh1Δ rnh201Δ* cells showed an extended chronological lifespan (Fig 5A). This result agrees with data from a budding yeast screen, where the *rnh201* mutant has been identified as long-lived [96]. Notably, in aged cells the double mutant showed an ~3-fold increase in chromosomal junctions on average, albeit with large variation, but no increase in indels (Fig 5B). Moreover, *rnh1Δ rnh201Δ* cells showed highly significant age-dependent accumulations of junctions at tRNAs, which are common sites of R-loop development, and at core Scw1 target genes (Fig 6F). These results further support the link between Scw1 and R-loop formation. Given the increased lifespan of *rnh1Δ rnh201Δ* cells (Fig 5A), these results also suggest that chromosomal junctions are not necessarily detrimental for longevity. It is possible that R-loop processing by RNase H, rather than the R-loops themselves, or indels, rather than junctions, are more detrimental in old cells.

We propose that diminished Scw1 function in old cells leads to exposed 3'-ends of nascent target mRNAs that can re-anneal with complementary sequences in their template DNA (S10 Fig). The resulting R-loops may then trigger genome instability and associated rearrangements downstream of the coding regions of Scw1 target genes. Thus, age-associated changes in an RBP with a role in post-transcriptional gene regulation can lead to non-random distribution of chromosomal rearrangements via increased genome instability at its targets.

## Conclusions

Our sensitive analyses of deeply sequenced yeast genomes uncover the rare and diverse events of chromosomal rearrangements that specifically accumulate during ageing of non-dividing cells. Our results provide evidence for widespread genome rearrangements in chronologically ageing fission yeast cells, associated with microhomology sequences near the breakpoints. Similar rearrangements are also evident in older human brain cells. How might DSBs be generated in non-dividing cells in the absence of replication? It has been shown that transcription-blocking topoisomerase I cleavage complexes, together with R-loop cleavage by endonucleases, can promote DSBs and cause neurological disorders [97]. Junctions indicative of rearrangements show a non-random distribution, both in terms of the regions that are joined (local DNA is often preferred to distal DNA) and the biological importance of the region (fewer junctions occur in coding regions). These rearrangements occur in global or local hotspots. Our findings point to different mechanisms acting at different types of hotspots and suggest that non-random patterns of DNA-RNA interactions play a prominent role, although other processes such

as transcription may contribute to the observed rearrangements. We find that the age-associated decline in the RNA regulatory protein Scw1 and associated R-loop formation can trigger breakpoint formation at target genes. Other RBPs that are modulated during ageing might act similarly to Scw1 in triggering genome rearrangements at other hotspots. These findings highlight that physiological changes can fuel cell- and condition-specific genome rearrangements with a non-random distribution.

## Materials and methods

### Strains used in this study

The ageing pools of yeast used for the main experiment are described elsewhere [34]. Briefly, an inter-crossed population fission yeast derived from the parental strains Y0036 and DY8531 [41,98,99] was inoculated into eight separate 2L flasks of liquid yeast extract supplemented (YES) medium, and grown until the optical density reached a plateau. Samples were then snap frozen in liquid nitrogen every 24hrs for the next five days. Scw1Δ and Scw1-TAP strains have been published elsewhere [85]. Samples for RIP-chip and ChIP-seq of Scw1-TAP were taken at 0, 2 and 4 days after cells reached a plateau in optical density. The Tlh2OE strain was generated for this study using a PCR-based approach [100] to introduce an *nmt1* promoter [101] in front of *tlh2*. Experiments using this strain, and experiments for chromosomal spreads, were carried out in Edinburgh minimal medium (EMM). For any lifespan experiments in EMM, cultures were grown until the optical density of cells reached a plateau, when cells were spun down and re-suspended in EMM without glucose. All transgenic strains were confirmed by PCR. The Sir2 deletion strain was obtained from the Bioneer fission yeast deletion library [102].

### DNA extraction, library preparation and sequencing

Genomic DNA was extracted using a standard phenol chloroform procedure [103]. After precipitation, DNA pellets were re-suspended in TE buffer and treated with RNase A (Qiagen), before being mechanically sheared to ~200bp (Covaris AFA). Sheared DNA was passed through PCR purification columns (QIAquick, Qiagen), and the fragment size distribution was checked using a 2100 Bioanalyzer (Agilent). Libraries were prepared using NEBNext Ultra kits (NEB) according to the manufacturer's standard protocol–this procedure included dA-tailing (S1 Text). After individual quantification (Qubit) and quality control (2100 Bioanalyzer), all forty-eight libraries were pooled. The pool was then sequenced using 126nt paired-end reads on the Illumina HiSeq 2500 (SickKids, Canada).

### Read alignment and junction filtration

After performing initial checks in FastQC [104], reads were aligned to the *S. pombe* reference genome (accessed May 2015) [59] using default parameters in BWA-MEM [105,106] (v0.7.12). Bam files were sorted, and PCR duplicates removed using Samtools [107] (v0.1.19). Split reads were then obtained using a simple shell script integrating Samtools [107] (v1.2). Note that, when assessing mitochondria-mitochondria split reads, the circularity of the mtDNA needed to be accounted for by ignoring those that aligned to both the start and end of the reference contig. Split reads were then filtered in a custom python script. To pass filtration, both alignments of each split read had to adhere to the following criteria: a minimum length of 40bp, a mapping quality score of 60 (the maximum score given by BWA-MEM), no clipped alignments at both ends [108]. Information on the alignments of each split read was obtained from the CIGAR strings of the initial soft-clipped read, as any information contained in subsequent hard-clipped reads is redundant. The 100bp sequencing reads from human data [24] were re-

mapped to the hg38n assembly of the human genome (accessed May 2016) using the same pipeline. The original purpose of this data set was to compare somatic mutations in young and old mitochondrial DNA, and the sequenced libraries were mtDNA-enriched [24]. However, there was still a considerable amount of coverage at the nuclear chromosomes, although less so than in the fission yeast data set.

To assess the pipeline, we generated one hundred random 100bp sequences of the *S. pombe* reference genome using BioPython [109] and output them as separate vcf files (the file format required by Mason [110]–see below). Each vcf file also contained a random location at which the fragment should be inserted. These fragments were then inserted into the reference genome to produce separate fasta files (using GATK [111] and Mason [110]). Each fasta file was used to generate 1x of simulated reads. Once all reads had been simulated, they were mixed with reads simulated using the standard reference genome (with no insertions) to give a proportion of split reads similar to that obtained in the real sequence files. The read alignment and junction filtration pipeline described above was then applied, and the number of simulated junctions that were recovered was counted. Although sensitivity was low (14/100 simulated rearrangements were recovered), there were no false positives, showing that this filtration is conservative but robust. Using a two-sample Wilcox test, a comparison between the GC-content of all simulated junctions ($N = 100$) and recovered junctions ($N = 18$) showed that there was no significant bias in our pipeline toward calling junctions in AT- or GC-rich regions ($W = 992$, $p = 0.5$).

## Modelling the expected rearrangement distribution

To calculate the expected number of DNA fusions between each chromosome in a random admixture of DNA ends, the number of reads mapping to each chromosome (all samples and timepoints considered simultaneously) was obtained using Samtools [107] (v1.2) and converted to a proportion of the total number of reads. In Perl, these proportions were used to calculate the number of junctions that would be expected for each combination of chromosomes, were the DNA ends to join randomly. One thousand simulations were performed.

## Measurement of microhomology at junctions

For each time point, bam files from all repeats were merged using Samtools (v1.2) [107]. Split reads were then obtained and filtered as above. Using python, junctions were first categorised as follows: those whose alignments share no sequence homology and have no non-homologous sequence between them; those whose alignments share no sequence homology and contain a non-homologous insertion between them; those whose alignments share an overlapping region of sequence homology (note that this homology is not necessarily perfect and may contain Indels). After categorising each junction, the length of any non-homologous or homologous sequence was recorded.

## Motif enrichment

For each junction classed as microhomology-mediated, we used BioPython [109] to collect the 100bp surrounding sequence. These 100bp sequences were compiled into a multi-sequence fasta file and submitted to MEME [112] for motif discovery. Only motifs with e-values lower than 0.05 were considered. Using TomTom [112], these sequences were compared to databases of known motifs for DNA-binding protein in fission and budding yeast. The most significant hits are shown in S5B Fig.

## Identification of rearrangement hotspots

Junction location files obtained after split read filtration were merged to create one combined file for each time point. A custom python script was used to divide each chromosome into 20kb windows and categorise each junction into a window based on its two alignments. The number of junctions at all windows was then output as separate chromosome-chromosome matrices for each time point. To see how much of an increase there was at each window, day five matrices were divided by their corresponding matrix at day zero (the end and start of the experiment, respectively). To quantify global hotspots, the average ratio across all windows at each 20kb bin of each chromosome was calculated.

## Coverage analysis

For the analysis in S6 Fig, the coverage at every position in the genome was obtained for each merged bam file using Bedtools [113] (v2.22.1). This per-base coverage was then used to obtain the median coverage at each 20kb bin. To get a score for how much each bin had changed in read depth, the median at day 5 for each bin was divided by the median at day zero.

## Junction intersection analyses

To see if junctions were enriched at coding regions, bed files for the locations of each feature were obtained. The coordinates of PomBase-annotated [114,115] coding regions were obtained from an *S. pombe* gff3 file (v31); the coordinates of Havana-annotated coding regions were obtained from a human gtf file (v85). For each set of real junctions analysed, an equally sized set of randomly located junctions was simulated using a custom python script. For example, in our yeast analyses there were eight repeats, which meant eight separate simulated sets for any comparison. Only nuclear junctions were used and simulated for all analyses. The intersection of each junction set with a given feature set was then made using Bedtools [113] (v2.22.1). The proportion of intersecting junctions in each real repeat was then compared to the proportion of intersecting junctions in each simulated repeat.

To analyse the distribution of junctions at the start and end of genes, a custom python script was used to collect CDS start and end positions based on their strand and coordinates from a bed file (see above). For each repeat, junction alignments were then collected at the 500bp surrounding these positions, and their relative positions inside or outside the gene end were recorded.

## Gene enrichment

To get the number of junctions in each 3' UTR, 3' UTR coordinates were obtained from an *S. pombe* gff3 file (see above) and converted into bed format. The number of junctions at each 3' UTR were then counted and plotted against the length of that UTR. Any gene with more junctions than twice the standard deviation of all UTRs were considered recurrently rearranged. Gene enrichment analysis was performed in AnGeLi [84].

## Chromosomal spreads and R-loop immunostaining

R-loop immunostaining and fluorescence quantification was performed as described [116], with the exception of cell lysis, where we mechanically broke cells in liquid nitrogen instead of using enzymatic lysis. R-loops were visualized using αS9.6 antibody [117]. For negative control samples, we added RNase H (Roche 10786357001) at 3u/100μl and incubated for two hours at 37˚C. Nuclei were stained with DAPI at 3μg/mL in 50% glycerol. Images were obtained using a spinning disk confocal microscope (Yokogawa CSU-X1 head mounted on Olympus body),

CoolSnap HQ2 camera (Roper Scientific), and a 100X Plan Apochromat 1.4 NA objective (Olympus). Images presented correspond to maximal projections of 10 slides' stacks using Image J open software [118].

## RIP-chip experiments

Two experimental repeats were performed for three timepoints: Days 0, 2 and 4 after cells reached stationary phase. RIP-chip of Scw1-TAP was performed as described [85], except for the following modifications: immunoprecipitation was carried out using monoclonal antibodies against protein A (Sigma); the lysis buffer contained 10mg/ml heparin (sigma H7405), 1 mM PMSF and 1:100 protease inhibitor cocktail (Sigma P8340); and magnetic beads containing the immunoprecipitate were resuspended in 50 µl of wash buffer containing 1 mM DTT, 1 unit/ml of SuperaseIN (Ambion 2696) and 30 units/ml of AcTev protease (Life Technologies 12575015). The solution with the beads was incubated for 1.5 h at 18˚C and the supernatant recovered, and RNA extracted using PureLink RNA micro columns (Life Technologies), according to the manufacturer's instructions. The RNA was eluted from the column in 14 µl and used for labelling without amplification. For microarrays, fluorescently labelled cDNA was prepared from total RNA and immunoprecipitated RNA from the RIP-chip using the SuperScript Plus Direct cDNA Labelling System (Life Technologies) as described by the manufacturer, except for the following modifications: 10 µg of total RNA was labelled in a reaction volume of 15 µl. We then used 0.5 µl of 10× nucleotide mix with labelled nucleotide (1/3 of the recommended amount), and added 1 µl of a home-made dNTP mix (0.5 mM dATP, 0.5 mM dCTP, 0.5 mM dGTP, 0.3 mM dTTP) to the reaction. All other components were used at the recommended concentrations. Note that these changes are essential to prevent dye-specific biases. Labelled cDNAs were hybridised to oligonucleotide microarrays manufactured by Agilent as described [36]. Microarrays were scanned with a GenePix 4000A microarray scanner and analysed with GenePix Pro 5.0 (Molecular Devices). A dye swap was performed for all repeats. Any gene with data missing at any probe or in either repeat were not included in the analysis. Data was median-centred for analysis and any genes whose expression was highly uncorrelated across repeats or did not appear at all timepoints were not considered.

## ChIP-seq experiments

ChIP-seq assays were performed as described [119]. For experiments using Scw1-TAP, ChIP-seq samples were collected at the same timepoints as for the RIP-chip experiments (see above). ChIP-seq libraries were sequenced using an Illumina MiSeq instrument. SAMtools [105] and BED tools [113] were used for sequence manipulation. Peak calling was performed using GEM [120].

## Single-molecule RNA Fluorescence *In Situ* Hybridization

We performed smFISH, imaging and quantification on formaldehyde-fixed cells as described [121]. Stellaris probes were designed and synthetized by Biosearch Technologies (Petaluma, CA). The *rpb1* probes were labelled with Quasar 670 and the *tlh2* probes with CAL Fluor Red 610. Probe sequences are provided in S1 Table. Cells were mounted in ProLong Gold antifade mountant with DAPI (Molecular Probes) and imaged on a Leica TCS Sp8, using a 63x/1.40 oil objective. Optical z sections were acquired (z-step size 0.3 microns) for each scan to cover the depth of the cells.

## DRIP-seq experiments

To detect R-loops, we snap-froze cells exponentially growing in YES. Samples for DRIP-seq were prepared based on published protocols [122–124], with minor variations. The main

difference was in the method for nucleic acid purification. Briefly, cells were thawed on ice and re-suspended in the same buffer used for ChIP-seq chromatin extraction. Cells were homogenized using a Fast-prep instrument, and chromatin was precipitated by centrifugation. Chromatin was re-suspended in Qiagen buffer G2 (from genomic extraction kit) with Proteinase K and incubated for 40 min at 55˚C. Nucleic acids were purified by phenol:chloroform extraction and precipitation, digested with S1 nuclease and sonicated. S1 nuclease prevents displacement of R-loops during sonication [122], while sonication has been reported to lead to better results than restriction-enzyme digestion [125].

As a control, half of the sample was treated with RNAse H in parallel, showing that RNase H led to reduced DRIP signals across the tRNA and rRNA genes (S11 Fig). R-loop-IP was performed using Dynabeads-mouse M280 pre-coated with S9.6 antibody (Millipore MABE1095). DNA was then eluted and purified for library preparation with NEBNext Ultra kits (NEB). The Bioanalyzer trace data showed that the fragments were about 400 bp after sonication, and sequencing libraries were selected to be 250–600 bp.

For data analysis, the depth at every base was collected using SAMtools and divided by the total number of mapped reads for each sample to normalise for differences in sequencing depth. To compare the R-loop signal at Scw1 target genes between wt and scw1Δ, we used a Python script to collect the mean normalised coverage for each target gene, including both those that were identified by Hasan and colleagues as upregulated in *scw1Δ* and those that were enriched in Scw1-TAP RIP-chip [85].

### Lifespans and sequencing of *sir2Δ* and *rnh1Δ rnh201Δ* strains

We used a recently developed assay to accurately measure lifespans of multiple yeast strains, simultaneously at a medium throughput [126]. Briefly, after waking strains up on YES plates, 20mL pre-cultures were grown in YES using 100mL flasks. These cultures were diluted to $OD_{600}$ = 0.002 in 160mL YES (in 500mL flasks) and grown to stationary phase. At each timepoint during chronological ageing, 100μL samples were taken to determine cell viability. Serial dilutions were made using a pipetting robot in a 96-well plate, and droplets were pinned to agar plates using the Singer RoToR. Colony forming units were then estimated using a maximum likelihood approach. At Day 0 and Day 3, 20mL samples were pelleted and frozen at -80˚C for DNA extraction. (At timepoints later than Day 3, too few live cells were present for sequencing analyses.) DNA extraction and library preparation were performed as described above. Genomic DNA was sequenced with 2x150bp reads on one lane of an Illumina MiSeq. Read alignment and junction filtration was then performed as described above.

For the indel analysis, all non-split reads containing insertions or deletions were collected using a shell script integrating Samtools. To calculate IPMR or JPMR, the number of indels or junctions in a given sample was divided by the number of mapped reads in the original bam file. To compare the relative IPMR or JPMR at Day 3 with that at Day 0, values were then divided by the median at Day 0 for each sample. For junction analysis at the *tlh2* global hotspot, junctions inside the hotspot were collected for wild-type and *sir2Δ* cells using a custom Python script. For junction analysis in *rnh1Δ rnh201Δ* cells, junctions were collected at selected features (tRNAs, Scw1 core targets) ±500bp using BedTools (version 2.27). JPMR was normalised to the mean JPMR at Day 0 for each genotype. Statistics were performed using the SciPy stats package.

### Supporting information

**S1 Fig. Examples of split reads.** Examples #1 and #2 show classic split reads with mapped mates, where a portion of the coloured read in Part 1 does not map anywhere near the read

which has been anchored to its location by its mate. A supplementary alignment of the unmapped part shows that it maps to a distal region of the genome (Part 2). Example #3 shows a situation where the mate of the read is not mapped (note that there is no line connecting either of the alignments in Part 1 or 2 to an anchoring read). Example #4 shows a more complex rearrangement between two nearby regions where the green portion (centre of the Part 1 panel, left of Part 2 panel) has been clipped and the supplementary alignment (purple) is mapped nearby in the same orientation. Depending on the relative location of each portion of the read, this type of event is suggestive of a deletion, translocation or repeat expansion/contraction. Note that in all examples, these reads are rare and our pipeline does not use the same levels of support required by most SV callers. Examples are snapshots from IGV.
(PDF)

**S2 Fig. Pipeline used for identification of breakpoint junctions.** Raw sequencing reads (126 nt paired-end, Illumina) were aligned to the reference genome with BWA-MEM. Alignment files were then converted to BAM format and sorted before having PCR duplicates removed in Samtools. Next, Bash commands (Awk and Grep) were used to extract split reads from these files. A custom Python script was used to filter these split reads to obtain a robust set of junctions representing two juxtaposed DNA locations, and to characterise their microhomology use across the junction.
(PDF)

**S3 Fig. Proportion of different junction types is not proportional to available DNA during library preparation.** Pie charts showing numbers of junctions between mitochondrial and nuclear (green), mitochondrial and mitochondrial (purple), and nuclear and nuclear (orange) DNA in simulated data (left, based on read depth at mitochondrial DNA and nuclear chromosomes) and in measured data (right).
(PDF)

**S4 Fig. Local DNA is preferred to distal DNA in intra-chromosomal rearrangements, but less so at global hotspots.** Kernel density and box plots showing the distribution of distances between juxtaposed pieces of DNA for any junction where one juxtaposed piece of DNA is at a global hotspot (bottom; light blue) and all other intra-chromosomal junctions at Day 5 (top; dark blue). These two groups were compared with a two-sample Wilcoxon rank sum test (p<0.0001).
(PDF)

**S5 Fig. Motifs at microhomology-mediated junctions.** A: All motifs discovered by MEME with e-values <0.05. This analysis was performed on all regions with microhomology-mediated junctions. Regions which aligned with known core sequence motifs of DNA-binding proteins in *S. cerevisiae* and *S. pombe* are underlined in grey and black, respectively (see B). B: Table showing the top hits for known *S. cerevisiae* (grey rows) and *S. pombe* (blank rows) DNA-binding protein motifs that align to the discovered motifs in A. An ID of the long motif from part A is detailed in the ID column, with the shorter aligning motif in the motif column.
(PDF)

**S6 Fig. Repeats are not responsible for extreme junction formation at hotspots.** A: Scheme showing how, in a model where repeats do not expand, the number of repeats at a locus should not affect the early:late ratio of junction numbers. B: Bee swarm plot showing the early:late ratio of read depth at each 20kb window, including the hotspots (red). A read depth ratio greater than 0 would indicate an increase in read depth (i.e. repeat expansion).
(PDF)

**S7 Fig. Transcriptional activity at the tlh2 locus may have a minor role in junction formation at global hotspots.** A: Left: Reads Per Kilobase Million (RPKM) of *tlh2* at the start of the ageing time course (100% cell viability) and when cell viability dropped to 50%. Two biological repeats of the time course were performed (each point corresponds to a repeat). Right: The same data as in Fig 4C are shown on a different scale to reveal the increased *tlh2* expression in older cells (50% viability). B: Left: Scheme for calculation of proportion of junctions downstream of *tlh2* (grey region; red crosses). Right: Results from three independent repeats with wild type (WT) and *tlh2* overexpression cells (tlh2OE). One-sided two sample Mann-Whitney U to test whether tlh2OE is greater than WT (U = 9, p = 0.04). C: Chronological lifespan of tlh2OE (red) compared to WT (grey). Lines show means of three repeats ± 68% confidence intervals.
(PDF)

**S8 Fig. During ageing, Scw1 binds less to some canonical RNA targets, and does not to bind DNA.** A: RIP-ChIP data showing log2-fold enrichment in Scw1 pull downs of all canonical target RNAs (81) with similar behaviour in both repeats and data at each time point. Two repeats were sampled at Day 0 (top), Day 2 (middle) and Day 4 (bottom rows). Clusters that show reduced enrichment with age are highlighted in purple. B: Example ChIP-seq of Scw1-TAP. Plots show normalised read depth (mean of two repeats) at three Scw1 targets (red) and three non-targets (blue) in cells aged for two days. For this visualisation, we chose genes which do not overlap any other annotations and matched targets with non-targets of similar gene and 3' UTR lengths.
(PDF)

**S9 Fig. R-loop immunostaining controls in exponential culture.** A: Representative images of nuclear spreads from wild-type and *rnh1Δ rn1201Δ* double mutant cells, with the latter known to feature increased R-loop formation [94]. Nuclei are stained with DAPI (blue) and R-loops are detected with the S9.6 antibody (green). To verify that the green signal comes from R-loops, control slides (right hand panels) were treated with the R-loop specific RNAse H before adding the primary antibody. Scale bar 10 μm. B: R-loop signal quantification for nuclear spreads shown in A. Chi-square: p <0.0001, N >50. C: It has been reported that in whole human cells most of the S9.6 signal arises from ribosomal RNA rather than R-loops, and that S9.6 signal remains unchanged by pretreatment with RNase H [95]. Our signal from isolated chromatin, however, was resistant to RNase III but sensitive to RNase H. Representative fields of nuclear spreads from wild-type and *scw1Δ* cells, each including untreated cells, cells pretreated with RNase III, and cells pre-treated with RNase H, as indicated. Scale bar 10 μm.
(PDF)

**S10 Fig. Model for how Scw1 levels might affect genome rearrangement at its targets.** In young cells, Scw1 is present and binds its targets at their 3' UTRs. In old cells, Scw1 is absent and thus leaves nascent RNA of its targets free to anneal with template ssDNA.
(PDF)

**S11 Fig. ChIP of S9.6 produces fewer reads at regions implicated in R-loop formation in RNase H-treated samples.** A: Normalised read depth at all tRNAs in wild-type samples. Blue: non-treated; green: RNase H-treated. B: Region of the *S. pombe* genome containing a tRNA and rRNA gene. Blue: non-treated; green: RNase H-treated.
(PDF)

**S1 Text. Arguments against breakpoint junctions forming *in vitro*.**
(PDF)

**S1 Table. Probes used for FISH experiment.**
(XLSX)

**S1 Data. Numerical data underlying graphs.**
(XLSX)

## Acknowledgments

We thank Caia Duncan and Juan Mata for Scw1 strains, Siôn L. Williams for providing human sequence data, Pawan Dhami (Genomics and Genome Engineering Facility funded by the Cancer Research UK-UCL Centre) for help with sequencing, and Mimoza Hoti for assistance with some of the sequencing runs.

## Author Contributions

**Conceptualization:** David A. Ellis, Jürg Bähler.

**Formal analysis:** David A. Ellis.

**Funding acquisition:** Samuel Marguerat, Jürg Bähler.

**Investigation:** David A. Ellis, Félix Reyes-Martín, María Rodríguez-López, Cristina Cotobal, Xi-Ming Sun, Quentin Saintain.

**Methodology:** David A. Ellis, Félix Reyes-Martín, María Rodríguez-López, Cristina Cotobal, Xi-Ming Sun, Samuel Marguerat, Víctor A. Tallada.

**Project administration:** Jürg Bähler.

**Resources:** Daniel C. Jeffares.

**Software:** David A. Ellis.

**Supervision:** Daniel C. Jeffares, Samuel Marguerat, Víctor A. Tallada, Jürg Bähler.

**Visualization:** David A. Ellis.

**Writing – original draft:** David A. Ellis, Jürg Bähler.

**Writing – review & editing:** David A. Ellis, María Rodríguez-López, Samuel Marguerat, Víctor A. Tallada, Jürg Bähler.

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
