## [Editor Report · Decision Letter 0]

24 Feb 2021

Dear Dr Bähler,

Thank you very much for submitting your Research Article entitled 'R-loops and regulatory changes in chronologically ageing fission yeast cells drive non-random patterns of genome rearrangements' to PLOS Genetics.

The manuscript was fully evaluated at the editorial level and by independent peer reviewers. The reviewers appreciated the attention to an important problem, but raised some substantial concerns about the current manuscript. Based on the reviews, we will not be able to accept this version of the manuscript, but we would be willing to review a much-revised version. We cannot, of course, promise publication at that time.

If you decide to revise the manuscript for further consideration at PLOS Genetics, please aim to resubmit within the next 60 days, unless it will take extra time to address the concerns of the reviewers, in which case we would appreciate an expected resubmission date by email to plosgenetics@plos.org.

[LINK]

We are sorry that we cannot be more positive about your manuscript at this stage. Please do not hesitate to contact us if you have any concerns or questions.

Yours sincerely,

Wolf-Dietrich Heyer

Guest Editor

PLOS Genetics

Wendy Bickmore

Section Editor: Epigenetics

PLOS Genetics

Dear Jürg

I trust you are doing well and coping with the pandemic.

Your manuscript was a very interesting read. The knowledge of the mechanism of genomic instability in quiescent and terminally differentiated cells is very limited, and your work constitutes a nice contribution. The reviews make a number of important points, and I find your clarifications and text changes/edition to be very helpful. Some of the technical issues with DRIP-seq would take a lot of refinement that the signal becomes completely RNAseH sensitive. Maybe some caveats could be acknowledged about this. The planned experiments with rnh1 rnh201 cells will be critical and one would predict a significant increase in split junctions. Wouldn’t it strengthen the argument to compare young and old quiescent wild type and mutant cells, to have the ageing effect? Also, the sir 2 experiment will make a nice addition, the same could be applied here. It is difficult to make a final decision before the outcome of these experiments is known. Especially, the rnh1 rnh201 experiment represents a critical test of the R-loop model. A negative result would argue against a causal relationship of R-loops and rearrangements and would require significant recasting of the manuscript.

I look forward to editing the revised version and good luck with the experiments.

Best regards

Wolf

---

## [Decision Letter · Decision Letter 1]

6 Aug 2021

Dear Jürg,

Thank you very much for submitting your revised Research Article entitled 'R-loops and regulatory changes in chronologically ageing fission yeast cells drive non-random patterns of genome rearrangements' to PLOS Genetics.

The manuscript was fully evaluated at the editorial level and by independent peer reviewers. The reviewers appreciated the attention to an important topic and the efforts in this revision. The reviews suggest a few  text changes and additions that I trust you have no problems addressing.

We therefore ask you to modify the manuscript according to the review recommendations. Your revisions should address the specific points made by each reviewer.

[LINK]

Yours sincerely,

Wolf-Dietrich Heyer

Guest Editor

PLOS Genetics

Wendy Bickmore

Section Editor: Epigenetics

PLOS Genetics

Reviewer's Responses to Questions

**Comments to the Authors:**

Reviewer #1: Ellis et al., present a really exciting and novel link between chronological aging, changes to RNA binding proteins and associated accumulation of R-loop linked DNA rearrangements. I appreciated the thoughtful revisions and rebuttal letter that the authors prepared for the previous reviewers. Overall, they seem to have addressed most of the concerns and have significantly improved the story. Here are a couple of points to consider clarifying in the manuscript.

- I appreciate that the authors have tempered their language comparing young and old brains in Figure 2. I would encourage them to be explicit in their language similar to the rebuttal letter. Basically stating the differences are marginally significant at best, and the really important finding is the same pattern of rearrangements in non-dividing somatic cells (or any age) and the chronologically aged S. pombe.

- On page 11 the authors cite Wang et al., to state that MMEJ proteins have been proposed to bind RNA:DNA hybrids. Which proteins, and are they conserved in S. pombe should be stated.

- On page 12/13 the identification of the Scw1 orthologues RBPMS and RBPMS2 in the search for the WRN gene is cited as somehow meaningful. I did not understand why this was included, surely this is a coincidence of mapping the WRN gene to a large region that happened to contain an RBPMS? Unless I am missing some context I would delete this statement.

- The Tlh1 hotspot is important, as are Scw1 target genes, but the authors do not tell us if the 3’ transcript at Tlh1 is a Scw1 target gene. Surely this can be extracted from their datasets and it should be mentioned one way or the other.

- The author acknowledge that the excess Tlh1 protein could be playing some role, since it's a RecQ helicase, but do not really explore this. Could Tlh1 protein overexpression, for example from a plasmid, cause a dominant genome instability phenotype and lead to rearrangements?

- I would make a point of citing the recent Chedin paper on S9.6 and highlighting the RNaseIII data, and the DRIP-seq data (which is less prone to artefacts), as a way to strengthen the conclusions. The immunofluorescence data with S9.6 cannot stand alone, but the authors have addressed this, they should make it clear to readers.

- Finally, it would be nice to hear more speculation about where the breaks come from at the tlh1 or other hot spots. In non-dividing cells replication fork collisions cannot be the driver. Sordet and Gromak had a paper a few years ago (PMID: 31533039) showing how nucleases and Topoisomerases can drive breaks at these sites. It would be worth citing and discussing whether you think something similar is happening.

Reviewer #2: No attachment.

On reading the rebuttal letter and the two versions of the manuscript I was able to appreciate the efforts and the quality of the improvements made in the revised version of the manuscript from Bahler and co-workers.

Considering the quality and originality of the initial observation, the wide array of methods accessible in yeast this work was positioned to generate solid and interesting results on DNA repair in ageing (stationary phase) cells. The sequencing and analysis methods used approach the very best in functional genomics, that was further improved in the revised version of the manuscript. The results are analyzed with a certain respect for possible caveats, which were rightly raised by the reviewers. The remarks of the reviewers in particular about hybrid RNA-DNAs allowed a relevant analysis of the mutants sir2, rnh1 and rnh201. The new results in the second version of the manuscript have made it possible to move from the description stage to the process stage allowing a model in which a subset of R-loops is favored in the absence of Scw1 and targets genome rearrangements. Personally, I am not a fan of this model, but it agrees with the results offered here.

Taken as a whole the new version of the manuscript is satisfactory for publication. However, I find the discussion genetically disappointing here, with the lack of discussion of the break-induce replication (BIR) process, Dicer's role in telomeres (tlh1-4) and rDNA (especially in fission yeast), topoisomerase 1, and more generally the organization of chromosomes in aging cells.

**Have all data underlying the figures and results presented in the manuscript been provided?**

Reviewer #1: Yes

Reviewer #2: Yes

PLOS authors have the option to publish the peer review history of their article (what does this mean?). If published, this will include your full peer review and any attached files.

Reviewer #1: No

Reviewer #2: No

---

## [Editor Report · Decision Letter 2]

18 Aug 2021

Dear Jürg,

We are pleased to inform you that your manuscript entitled "R-loops and regulatory changes in chronologically ageing fission yeast cells drive non-random patterns of genome rearrangements" has been editorially accepted for publication in PLOS Genetics. Congratulations!

The revised manuscript adequately addressed the remaining concerns and will make a valuable contribution to the field.

Yours sincerely,

Wolf-Dietrich Heyer

Guest Editor

PLOS Genetics

Wendy Bickmore

Section Editor: Epigenetics

PLOS Genetics

Comments from the reviewers (if applicable):

**Data Deposition**

http://datadryad.org/submit?journalID=pgenetics&manu=PGENETICS-D-21-00147R2

**Press Queries**

---

## [Editor Report · Acceptance letter]

25 Aug 2021

PGENETICS-D-21-00147R2 

R-loops and regulatory changes in chronologically ageing fission yeast cells drive non-random patterns of genome rearrangements 

Dear Dr Bähler, 

We are pleased to inform you that your manuscript entitled "R-loops and regulatory changes in chronologically ageing fission yeast cells drive non-random patterns of genome rearrangements" has been formally accepted for publication in PLOS Genetics! Your manuscript is now with our production department and you will be notified of the publication date in due course.

With kind regards,

Zsofi Zombor

PLOS Genetics

On behalf of:
